# An Early Indicator in Evaluating Cardiac Dysfunction Related to Premature Ventricular Complexes: Cardiorespiratory Capacity

**DOI:** 10.3390/healthcare11222940

**Published:** 2023-11-10

**Authors:** Xiaozhu Ma, Jiangtao Yan, Wanjun Liu

**Affiliations:** Division of Cardiology, Department of Internal Medicine, Tongji Hospital, Tongji Medical College, Huazhong University of Science & Technology, Wuhan 430030, China; xzma2023@hust.edu.cn (X.M.); jtyan@tjh.tjmu.edu.cn (J.Y.)

**Keywords:** premature ventricular complexes, cardiac capacity, cardiopulmonary exercise test, oxygen uptake

## Abstract

Cardiac dysfunction induced by premature ventricular complexes (PVCs) is relatively controversial and challenging to detect in the early stage. In this observational study, we retrospectively analyzed the cardiopulmonary exercise test (CPET) data of 94 patients with frequent premature ventricular beats (47 males, 49.83 ± 13.63 years) and 98 participants (55 males, 50.84 ± 9.41 years) whose age and gender were matched with the patient with PVCs. The baseline information and routine echocardiography detection were recorded on admission. PVCs were diagnosed by 24 h Holter monitoring, and cardiorespiratory capacity was assessed using peak oxygen uptake (V’O_2_peak), anaerobic threshold (AT), and other CPET parameters with an individualized bicycle ramp protocol according to the predicted workload and exercise situation of each participant. There were no statistically significant differences in most baseline characteristics between the two groups. Indicators that reflect cardiopulmonary capacity, such as V’O_2_peak, AT, and ΔO2 pulse/Δwork rate(ΔV’O_2_/ΔWR), were all significantly lower in the PVC group (*p* = 0.031, 0.021, and 0.013, respectively) despite normal and nondiscriminatory left ventricular ejection fractions between the two groups. However, there was no statistically significant difference among subgroups based on the frequency of PVCs, which was <10,000 beats/24 h, 10,000–20,000 beats/24 h, and >20,000 beats/24 h. The cardiorespiratory capacity was lower in patients with frequent PVCs, indicating that CPET could detect early signs of impaired cardiac function induced by PVCs.

## 1. Introduction

Premature ventricular complexes (PVCs) refer to early depolarization of the ventricles with or without mechanical contraction and are one of the most common types of arrhythmia [1]. With the widespread availability of diagnostic technologies, it is not surprising that an increasing number of patients with PVCs have been timely diagnosed in recent years. Initially, PVCs were considered benign without cardiac structural alterations. However, several research studies have revealed that an increasing burden of PVCs is associated with reduced left ventricular systolic function [2,3,4] and higher short-term mortality [5]. Apart from reduced cardiac function, a special kind of cardiomyopathy, namely PVC-induced cardiomyopathy, may also develop due to long-term burden of PVCs [6]. Unfortunately, it is challenging to detect these adverse effects on the cardiac function in the early stage of arrhythmia given that patients in the early stages of the disease usually do not exhibit obvious clinical symptoms.

Cardiopulmonary exercise testing (CPET) is a dynamic and noninvasive clinical test that provides an integrative and objective evaluation of multiple organs and systems, particularly pulmonary and cardiovascular function under physiological stress. It appears to be a promising method for assessing cardiac function before the detection of virtual structural and functional damages. CPET was initially proposed as a useful tool for grading the severity of heart failure in the 1980s [7] and has since been utilized in the management of various cardiovascular diseases. According to the latest 2021 European Society of Cardiology (ESC) Guidelines [8] for the diagnosis and treatment of acute and chronic heart failure, a cardiopulmonary exercise test is recommended as part of the evaluation and management of heart transplantation and/or mechanical circulatory support, as it helps to optimize the prescription of exercise training. However, the use of CPET in current clinical practice primarily focuses on heart failure and ischemic heart disease, while its application to evaluate the effects of PVCs on cardiorespiratory capacity is uncommon.

Therefore, the purpose of this study was to identify early cardiopulmonary impairment caused by PVCs and explore the advantage of CPET in evaluating cardiorespiratory performance in real-world patient care.

## 2. Materials and Methods

### 2.1. Population

This was a retrospective cohort study conducted at Tongji Hospital, affiliated with Tongji Medical College, Huazhong University of Science and Technology. Patients with symptoms such as palpitation, chest pain, chest tightness, or abnormal electrocardiography findings, who had undergone 24 h Holter monitoring and consented to CPET, were included in this study. Thereinto, patients with more than 1000 PVCs in the 24 h Holter monitoring and CPET performed within three days of the monitoring were classified as the PVC group. The exclusion criteria were as follows: (1) apparent wheezing and dyspnea; (2) severe arrhythmia; (3) acute coronary syndrome within the past three months; (4) chronic heart failure; (5) pregnancy; (6) uncontrolled severe obstructive lung disease; (7) severe hepatic or kidney dysfunction; and (8) moderate to severe aortic and mitral stenosis. Ninety-four patients with PVCs met these criteria and were included in the subsequent analysis. Simultaneously, patients who were age-, gender-, and body mass index (BMI)-matched with the PVC groups were enrolled as controls. Although controls had been referred to our hospital for similar reasons, their PVC frequency was less than 1000 times per 24 h.

The demographic and clinical information, including age, height, weight, BMI, cardiovascular risk factors, and medication use, presented in Table 1, were collected from the Tongji Hospital information system. Echocardiography was performed by two professional sonographers according to standard protocol. The routine parameters reflecting cardiac systolic function (left ventricular ejection fraction and fractional shortening) and diastolic function (E/E′ and E/A value) were recorded. In addition, the morphology features of PVC beats, including the QRS duration, pseudo delta waves, the QRS axis in limb leads, and bundle branch-like morphology were evaluated by two professional cardiologists. This research study was approved by the ethics committee of the institutional review board of Tongji Hospital and written informed consent was obtained from all participants.

### 2.2. CPET Method

All subjects had been referred for CPET at our hospital settings to estimate their cardiopulmonary function. The participants were evaluated for indications by a professional physician and signed an informed consent form regarding relevant risks and benefits before CPET. The participants were instructed to abstain from smoking for at least 8 h, drinking coffee, and eating a full meal. Oxygen consumption, carbon dioxide production, ventilatory capacity, and hemodynamic indices were continuously measured throughout the test using professional equipment (CARDIOVIT CS-200 Excellence). After the preliminary assessment of static lung function, the participants were assigned to complete symptom-limited maximum extreme exercise testing on a cycle ergometer with a personalized ramp exercise protocol until exhaustion. Specifically, we recorded the initial resting state data for three minutes, and then the participants started warm-up cycling without load at a pedal speed of 60 r/min for three minutes. The increasing power rate was set to 10–30 W/min based on the patients’ age, sex, height, usual exercise habits, and estimated functional state. Patients reached the maximum exercise limit with symptom restriction within 6–10 min and their recovery was recorded for more than five minutes. The prespecified criteria for interrupting the exercise test were as follows: (1) adverse clinical symptoms, such as fatigue, dyspnea, severe chest pain, dizziness, faintness, sudden pallor, and loss of coordination; (2) abnormal ECG presentation, such as pathological Q waves, severe arrhythmia, ≥2 mm ST depression or ≥1 mm ST elevation in at least two adjacent leads; (3) respiratory exchange ratio (RER) ≥ 1.15; and (4) upon request of the participant.

A large number of variables were typically measured. V’O_2_ refers to the ability of the human body to inhale and utilize oxygen during physical activity. V’O_2_peak, which describes the highest achieved V’O_2_ value during an exercise test, is the most important parameter to reflect cardiopulmonary capacity. Here, we defined V’O_2_peak as the average value of the last ten seconds of CPET. V’CO_2_ is used to quantify the amount of carbon dioxide produced or exhaled by an individual, reflecting metabolic state during exercise. As an alternative to V’O_2_peak in determining exercise tolerance, the anaerobic threshold (AT) was identified using a V-slope analysis of V’O_2_ and V’CO_2_ and was confirmed by specific trends of ventilatory equivalent for oxygen (VE/V’O_2_), ventilatory equivalent for carbon dioxide (VE/V’CO_2_), end-tidal pressure of oxygen, and end-tidal pressure of CO_2_. Other routine variables, including V’O_2_AT as a percentage of the predicted V’O_2_, VE/V’CO_2_ slope, peak O_2_ pulse, the slope of change in V’O_2_ to change in work rate, V’CO_2_, VE, HR, blood pressure at peak, and AT were recorded synchronously. The Borg scale (6–20 scale) which allowed individuals to subjectively rate their level of exertion during exercise, was performed after the test.

### 2.3. Statistical Analysis

All statistical analyses were conducted using IBM SPSS Statistics for Windows, version 29.0 (IBM Corp., Armonk, NY, USA) [9]. Numerical variables were expressed as mean ± standard deviation, while categorical variables were presented as counts (percentage). The normality of distribution for the data was assessed by the Kolmogorov–Smirnov test. Analysis on continuous variable data between the PVCs and control groups were evaluated by the unpaired t-test if the data was normally distributed. Otherwise, the comparison between the two groups would be conducted using the Mann–Whitney U test, such as NTpro-BNP. For the comparison of categorical variables, the Chi-square test was performed. Statistical significance was defined as *p* < 0.05. Effect size was expressed by the Cohen d whose absolute value greater than 0.2 representing the obvious difference between control and PVC groups. Three subgroup analyses were performed using univariate ANOVA followed by post hoc analysis using LSD Test. The correlation between PVC burden and related CPET parameters was assessed using Pearson’s test. It was considered a strong correlation when the absolute value of r was greater than 0.6.

## 3. Results

### 3.1. Baseline Information

For the PVC group, 48 patients (51%) visited the hospital due to symptoms of palpitation or chest discomfort, while the remaining 46 cases (49%) were incidentally diagnosed as PVCs during a medical examination (Figure 1). Table 1 presents the specific baseline information of the PVCs and control groups. The basic demographic data, including age and gender (*p* > 0.05), showed no significant differences between the groups. There were also no statistically significant differences in the risk factors for cardiovascular disease, including hypertension, hyperlipidemia, diabetes, and diseases that may affect the cardiopulmonary capacity. A specific classification based on the severity of coronary heart disease indicated there were no significant differences in the degree of coronary artery stenosis between two groups. We also compared the medication history between the two groups, and our findings underlined that the use of beta blockers, which was highly related to heart rate and cardiac function and tended to be similar (*p* = 0.204).

Table 2 showed the morphological features of arrhythmia. In addition, as shown in Table 3, there were no significant differences in ventricular wall thickness (interventricular septum, *p* = 0.074; left ventricular posterior wall, *p* = 0.305) and functional changes detected by echocardiography between the two groups.

### 3.2. CPET Data

Of the 94 PVCs patients and 98 controls enrolled based on the aforementioned inclusion and exclusion criteria, none of them terminated the exercise test due to severe clinical symptoms or abnormal ECG presentation. The Borg scale score of all participants was greater than 13. At the same time, suppression of frequent PVCs was noted in all patients. A comparison of CPET variables between groups was shown in Table 4 and Figure 2. Firstly, no significant differences were seen regarding the heart rate and blood pressure at rest. However, there were significant differences in V’O_2_peak/kg (20.92 ± 4.99 in the control group vs. 19.32 ± 5.20 in the PVC group, *p* = 0.031) and V’O_2_AT/kg (13.36 ± 2.70 in the control group vs. 12.40 ± 3.02 in the PVC group, *p* = 0.021). To our surprise, the important parameters representing impaired oxygen delivery (V’O_2_peak (%pred), V’O_2_AT(%pred), and ΔV’O_2_/ΔWR) also decreased (*p* = 0.009, 0.005, and 0.013, respectively). No significant differences were found in heart rate at peak exercise between the two groups. We further divided the PVC patients into three groups based on the frequency of PVCs, namely < 10,000 beats/24 h, 10,000–20,000 beats/24 h, and >20,000 beats/24 h. Interestingly, significant difference was noted in V’O_2_/KG peak but not in other main parameters among the three subgroups, which was shown in Table 5. Additionally, there was no significant correlation between the burden of PVCs and V’O_2_peak (Figure 3). In our cohort, the origin of PVCs in most patients was the right ventricular outflow tract (80.7%).

## 4. Discussion

In the present study, we identified that patients with frequent PVCs had lower V’O_2_ peak compared to the controls, indicating a reduction in cardiopulmonary reserve function in patients with frequent PVCs and normal cardiac structure.

Premature ventricular complexes are common type of arrhythmia in clinical practice and frequent PVCs are considered risk factors for left ventricular dysfunction through long-term follow up observations. A previous cohort study on over 1100 participants without congestive heart failure (CHF) revealed that the population-level risk for incident heart failure attributed to PVCs was 8.1% [4]. Meanwhile, a portion of PVCs patients developed PVC-induced cardiomyopathy, which refers to a reversible reduction in left ventricular systolic function. Several research groups have indicated that LV dysfunction in patients with frequent PVCs could be improved to some extent after catheter ablation therapy [10,11]. However, the effects on cardiac function are not comparable in the early stages of frequent PVCs. In a previous study, most patients did not exhibit any clinical symptoms of reduced heart function, such as shortness of breath after exertion, and more than half of all patients failed to demonstrate any evident decrease in left ventricular ejection fraction (LVEF) [12]. Arrhythmogenic cardiomyopathy (ACM) is a myocardial disorder that can lead to sudden cardiac death. Ventricular arrhythmias may be an early phenotypic expression in ACM patients, characterized by PVCs which are inconsistent with the degree of cardiac dysfunction [13]. In general, patients with fewer symptoms are thought to have a benign prognosis. However, researchers have found that many asymptomatic patients were usually overlooked. As a result, they were more susceptible to long-term PVC burden and progression to PVC-induced cardiomyopathy [14]. Despite the significant decrease in LVEF among patients with highly frequent PVCs, the prognosis was found to be worse in patients with exercise-induced PVCs but without cardiac structural changes after long-term follow-up [15]. In our study, we were surprised to identify reduced cardiac capacity induced by PVCs prior to observing a decrease in routine echocardiography parameters. In our study, there were some differences in NT-proBNP levels between the two groups. However, its value was influenced by several factors, such as age, gender, renal function, and coronary artery disease. Of note, two cases in our cohort had significantly increased NT-proBNP values due to poor renal function. Consequently, NT-proBNP level could not reflect differences in cardiac function between the groups. Furthermore, the value of E/E′ in the PVC group was higher than in the controls group, but still within the normal range, which seemed to mean that there was impaired left ventricular diastolic function in the early stage. These further confirmed the special potential of CPET in identifying early cardiac dysfunction which is difficult to detect with routine echocardiography tests. Hence, this will enable the development of early prevention strategies and aggressive therapies of the underlying disease.

The mechanism of LV dysfunction observed in patients with frequent PVCs is not completely elucidated. Controversy still exists between findings in laboratory animals and human studies. Tachycardia-induced cardiac functional decline might be one of the important factors attributed to the progression of LV dysfunction [16] although the effect is probably nonessential. Nevertheless, bradycardia is also a potential confounder because cardiac output decreases with low heart rates. The concept of “PVCs-induced cardiomyopathy” has recently gained a lot of attention, and changes in cardiac rhythm do not appear to provide a complete explanation for this phenomenon. Left ventricular dyssynchrony is another plausible contributing factor. Long-term or high frequency PVCs alter the normal hemodynamic status, leading to increases in LV filling pressure and LA overload. As a result, the volume of blood pumped into peripheral tissues is insufficient to meet the metabolic demands of tissues. A study performed by Tomos [17] revealed the association between severe left ventricular dyssynchrony and PVCs-induced cardiomyopathy in an animal model. In addition, it is likely that some occult structural heart disease associated with the etiology of PVCs may have existed and was not detected by our conventional monitoring methods. Modifications in cardiac anatomy resulting from PVCs of the left ventricular outflow tract origin were confirmed. In our study, PVCs mainly originated on the right ventricle, but there was no statistically significant difference in V’O_2_peak when dividing patients according to the site of PVCs origin. However, the classification of sites could have been refined further, allowing the identification of the precise origin and differences in cardiopulmonary function caused by ventricular premature complexes of different origins.

Genetic differences should be considered as a factor in the development of arrhythmia as it is challenging to explain the different responses noted among patients using the same treatment parameters. In a meta-analysis [18], genome-wide association studies were carried out and researchers identified that mutations in a certain gene locus contributed to the differences in the risk of morbidity due to supraventricular and ventricular ectopy. ACM is perceived to be highly correlated with genetic background. Consequently, this comes to underline that we need to be very careful when identifying potential cardiomyopathy through cardiac magnetic resonance and genetic testing and be hypervigilant against patients whose cardiac capacity has been impaired as detected by CPET.

The difference among the three subgroups classified by PVCs frequency indicates that the effect of varying frequency may not be apparent in the early stage of the disease. While there was significant difference in V’O_2_/KG peak in patients with less than 10,000 PVC beats per 24 h as compared to patients with more than 20,000 PVC beats. However, it should be noted that PVCs-induced cardiac impairment is not solely attributable to its magnitude. Multiple factors, such as PVCs location, morphology, and QRS duration [19], may exert significant impacts. We believe that the correlation between the burden of PVCs and important indexes, which showed a negative trend but no statistical significance, could be ascribed to non-persistent arrhythmia and the relatively small sample size included in this study.

Moreover, identical findings were noted when the frequency of PVCs was suppressed by exercise during CPET in all patients, which was generally interpreted as an adrenergic response. A previous study demonstrated that PVCs were suppressed either during the recovery phase of exercise testing or after exercise testing during a mean follow up of 7.2 years in children [20], and this suppression was associated with a benign prognosis. In contrast, exercise-induced PVCs have been linked to death and the occurrence of cardiovascular events in adults [21]. Changes in electrocardiograms during CPET can be a useful method to assess the presence or absence of underlying ischemic heart disease or arrhythmia. In our study, PVCs were suppressed during exercise in all participants. Further investigation into the specific mechanism and different prognostic outcomes needs to be conducted in large sample studies.

## 5. Conclusions

In conclusion, we discovered patients with frequent PVCs have worse cardiorespiratory capacity than the controls, and further investigation on the association between PVCs burden and cardiorespiratory capacity is necessary to excavate the important role of CPET in disease diagnosis.

## 6. Limitations

The current study has several limitations. Firstly, this is an observational study with a relatively small sample size. Large long-term follow-up clinical trials are required to confirm and generalize our results in the future. Secondly, we mainly focused on cardiopulmonary capacity between patients with PVCs and those without PVCs. However, the large frequency variations within PVC group could not be avoided. Considering the research type, our conclusion has limited guidance for clinical decision-making in real clinical practice. Thirdly, global longitudinal strain is an appropriate parameter to evaluate subtle LV function changes. Unfortunately, it is not currently routine parameter of echocardiogram in our hospitals and is therefore not available in this observational study. Furthermore, inter-individual variation in exercise capacity should not be neglected, as individual differences in lifestyle may exert an impact on the CPET results, particularly in the case of a small sample size. In addition, the ECG recording times of some participants were not accurate for 24 h. Therefore, the comparison among patients with different frequencies of PVCs needs further verification.

## Figures and Tables

**Figure 1 healthcare-11-02940-f001:**
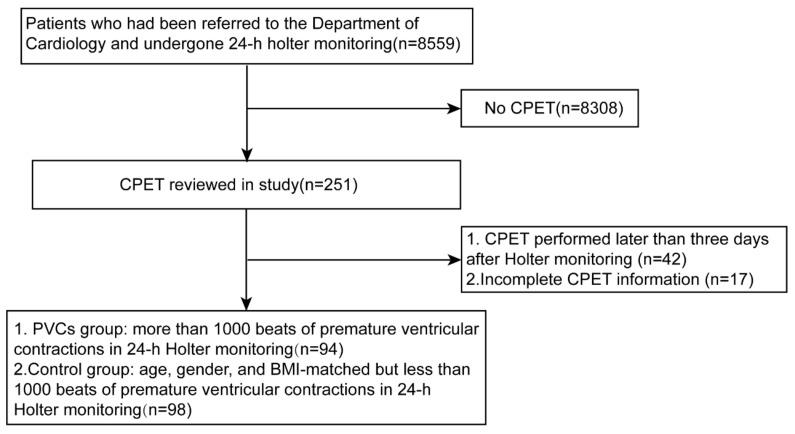
Flow diagram showing inclusion of participants in the study. CPET, cardiopulmonary exercise test; PVCs, premature ventricular complexes.

**Figure 2 healthcare-11-02940-f002:**
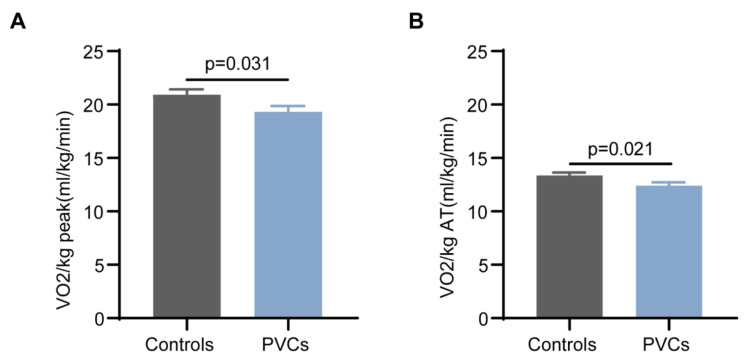
Weight-adjusted peak oxygen consumption between control group and PVC group. The comparison about VO_2_/kg at peak (**A**) and anaerobic threshold (**B**) between the two groups. Abbreviations: V’O_2_, oxygen consumption; AT, anaerobic threshold.

**Figure 3 healthcare-11-02940-f003:**
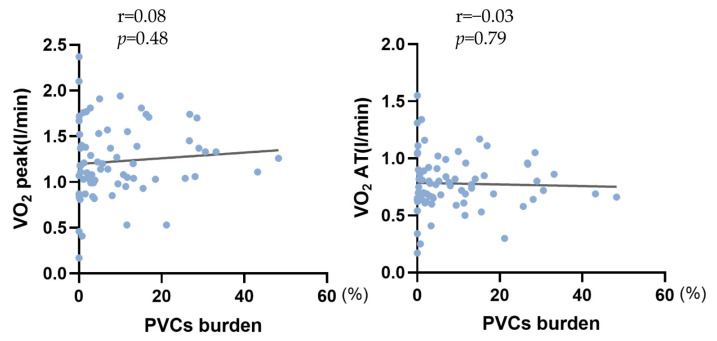
Correlation between PVCs burden and main parameters of CPET. Abbreviations: V’O_2_, oxygen consumption; AT, anaerobic threshold. Statistical significance was defined as *p* < 0.05.

**Table 1 healthcare-11-02940-t001:** Baseline characteristics of the study population.

Subject	Controls (n = 98)	PVCs (n = 94)	*p* Value
Age (yrs)	50.84 ± 9.41	49.83 ±13.63	0.551
Male (n,%)	55 (56.1)	47 (50.0)	0.395
Height (cm)	166.54 ± 7.58	165.97 ± 6.89	0.590
Weight (kg)	69.39 ± 13.38	65.92 ± 10.95	0.052
Body mass index (kg/m^2^)	24.89 ± 3.65	23.87 ± 3.36	0.050
Smoking (n,%)	15 (15.3)	11 (11.8)	0.484
CAD (n,%)	18 (18.1)	16 (17.0)	0.156
ACS (n,%)	10 (10.2)	9 (9.6)	0.892
Number of lesional coronary artery			0.771
1-vessel CAD (n,%)	6 (6.1)	8 (8.5)	
2-vessel CAD (n,%)	4 (4.1)	3 (3.1)	
3-vessel CAD (n,%)	8 (8.2)	5 (5.3)	
Gensini score	16.77 ± 11.45	19.20 ± 17.01	0.672
Hypertension (n,%)	33 (33.7)	47 (50.0)	0.071
Hyperlipidemia (n,%)	23 (23.5)	24 (25.5)	0.997
Diabetes (n,%)	12 (12.2)	14 (14.9)	0.758
Medication use			
Beta-blocker (n,%)	44 (44.9)	37 (37.8)	0.204
Calcium channel blocker (n,%)	19 (19.4)	28 (28.6)	0.178
Laboratories			
NT-proBNP (pg/mL)	35.5 (49.5)	62.1 (83.0)	0.044 ^†^
cTnI (ng/mL)	5.32 ± 14.30	4.24 ± 6.81	0.509
eGFR (mL/min/1.73 m²)	91.86 ± 12.83	89.34 ± 20.97	0.532

^†^ Comparison between the two groups conducted using the Mann–Whitney U test. Abbreviations: PVC, premature ventricular complex; CAD, coronary artery disease; ACS, acute coronary syndrome; NT-proBNP, N-terminal pro-brain natriuretic peptide.

**Table 2 healthcare-11-02940-t002:** Morphological features of PVCs beats.

Characteristics	
QRS duration (ms)	140 ± 33
pseudo delta waves (n,%)	10 (10.6)
axis	
superior axis (n,%)	24 (25.5)
inferior axis (n,%)	70 (74.5)
bundle branch-like morphology	
left (n,%)	77 (81.9)
right (n,%)	17 (18.1)

**Table 3 healthcare-11-02940-t003:** Echocardiographic indices of study patients.

Subject	Controls	PVCs	*p* Value
LV size (mm)	47.02 ± 4.68	47.93 ± 4.83	0.212
LA size (mm)	35.62 ± 4.99	35.88 ± 5.78	0.749
IVS(mm)	9.93 ± 1.80	9.44 ± 1.80	0.074
LVPW (mm)	9.46 ± 1.32	9.24 ± 1.52	0.305
LVEF (%)	63.11 ± 6.08	60.96 ± 8.1	0.055
FS (%)	34.47 ± 5.32	33.23 ± 6.19	0.157
E/A	1.17 ± 0.53	1.27 ± 0.58	0.224
E/E′	11.46 ± 3.61	12.6 ± 5.45	0.107

Abbreviations: LV, left ventricle; LA, left atrium; IVS, interventricular septum; LVPW, left ventricular posterior wall; LVEF, left ventricular ejection fraction; FS, fractional shortening.

**Table 4 healthcare-11-02940-t004:** CPET parameters in PVCs and controls.

Subject	Controls (n = 98)	PVCs (n = 94)	*p* Value	Reference Value	Effect Size
AT (l/min)	0.91 ± 0.22	0.80 ± 0.24	0.002	-	−0.478
V’O_2_ peak (l/min)	1.42 ± 0.39	1.25 ± 0.40	0.004	-	−0.430
V’E peak (l/min)	48.26 ± 12.16	42.37 ± 12.29	0.001	-	−0.482
SBP rest (mmHg)	129.74 ± 26.15	126.61 ± 21.74	0.370	-	−0.130
DBP rest (mmHg)	78.14 ± 15.64	78.52 ± 13.70	0.859	-	0.026
HR rest (bpm)	87.40 ±13.81	87.96 ± 14.63	0.785	-	0.039
HR AT (bpm)	111.39 ± 14.97	108.18 ± 18.50	0.188	-	−0.191
HR peak (bpm)	142.66 ± 17.88	138.88 ± 26.58	0.247	-	−0.168
Load AT (W)	59.34 ± 21.06	52.49 ± 23.51	0.035	-	−0.307
Load peak (W)	101.97 ± 37.97	92.53 ± 37.35	0.084	-	−0.251
V’O_2_/HR peak	0.010 ± 0.003	0.009 ± 0.003	0.016	-	−0.333
V’O_2_ peak (%pred)	0.74 ± 0.14	0.68 ± 0.15	0.009	≥85% pred	−0.414
V’O_2_ AT (%pred)	0.48 ± 0.11	0.44 ± 0.10	0.005	40–60%	−0.380
V’E/V’CO_2_ slope	26.44 ± 5.18	26.76 ± 7.58	0.750	<35	0.049
V’O_2_/KG AT (ml/kg/min)	13.36 ± 2.70	12.40 ± 3.02	0.021	-	−0.336
V’O_2_/KG peak (ml/kg/min)	20.92 ± 4.99	19.32 ± 5.20	0.031	-	−0.314
V’E/KG peak (ml/kg/min)	713.62 ± 155.25	657.38 ± 168.60	0.017	-	−0.347
ΔV’O_2_/ΔWR ((ml/min)/W)	9.69 ± 1.92	8.84 ± 2.66	0.013	>8.6	−0.368

Abbreviations: PVCs, premature ventricular complexes; V’O_2_, oxygen consumption; AT, anaerobic threshold; SBP, systolic blood pressure; DBP, diastolic blood pressure; V’CO_2_, carbon dioxide output; V’E, ventilation; HR, heart rate; WR, work rate.

**Table 5 healthcare-11-02940-t005:** Comparison of main parameters among different subgroups *.

						Post Hoc	
Subject	<10,000/24 h (n = 55)	10,000–20,000/24 h (n = 23)	>20,000/24 h (n = 16)	*p* value	<10,000/24 h vs. >20,000/24 h	10,000–20,000/24 h vs. >20,000/24 h	<10,000/24 h vs. 10,000–20,000/24 h
AT (l/min)	0.78 ± 0.25	0.88 ± 0.25	0.79 ± 0.21	0.270			
V’O_2_ peak (l/min)	1.18 ± 0.41	1.36 ± 0.36	1.35 ± 0.36	0.110			
V’O_2_ AT (%pred)	0.45 ± 0.11	0.43 ± 0.1	0.40 ± 0.08	0.289			
V’O_2_ peak (%pred)	0.67 ± 0.16	0.66 ± 0.12	0.70 ± 0.15	0.726			
V’O_2_/KG AT (ml/kg/min)	12.01 ± 3.03	13.23 ± 3.61	12.7 ± 2.38	0.282			
V’O_2_/KG peak (ml/kg/min)	18.18 ± 5.29	20.46 ± 5.31	21.67 ± 4.34	0.040	0.023	0.490	0.090

* Groups compared with univariate ANOVA (LSD post hoc test). Abbreviations: V’O_2_, oxygen consumption; AT, anaerobic threshold.

## Data Availability

Data are contained within the article.

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
