# Peer review of "An Early Indicator in Evaluating Cardiac Dysfunction Related to Premature Ventricular Complexes: Cardiorespiratory Capacity"

_healthcare, 2023, doi:10.3390/healthcare11222940_

Round 1

Reviewer 1 Report

Comments and Suggestions for Authors

This study examined the usefulness of the cardiopulmonary exercise test (CPET) in patients with symptoms of PVCs. This is an interesting study for clinicians, and I think that CPET, a physiological test, may be able to estimate cardiac reserve.

However, I would like the authors to correct the following points.

Major revision

1) The authors considered two groups, but in defining the PVC group, it was assumed that there were more than 1000 PVCs. The authors must explain why the number of PVCs in the PVC group was set to more than 1000.

2) The PVC group has been shown to have a lower V'O2 peak than the control group. However, it would seem that the duration of illness since the onset of symptoms associated with PVCs would be an issue in this regard. If the duration of illness of the patient is known, it would be good to state it, but I would like to hear the authors' thoughts on this.

Minor revision

1) In the abstract (Line 10), CPET suddenly appears; please spell it out.

2) V'O2 and V'CO2 may also need to be spelled out. Please correct the text and provide notes in the Figure.

Comments on the Quality of English Language

None

Author Response

Thank you very much for taking the time to review this manuscript. we have revised the original manuscript. The revised parts are shown in the re-submitted files with red font. Please see the attachment about the point-by-point response.

Reviewer 2 Report

Comments and Suggestions for Authors

In this manuscript the authors evaluated the role of cardiopulmonary exercise test (CPET) in patients with premature ventricular complexes (PVC) and cardiac dysfunction.

I would like to underline some major issues related to this work.

MAJOR ISSUES

·       There is an ongoing intense debate over the definition of “PVC-induced cardiomyopathy”. Over the last years, significant progresses in the field of advanced imaging (especially CMR) and genetics have revealed the presence of concealed cardiomyopathies and genetic disorders in these patients. The authors did not perform CMR or extensive genetic analysis in this cohort. Therefore, they could not exclude the presence of specific cardiomyopathies in these patients. Moreover, some specific variants (dystrophin, lamin A/C, phospholamban) have been associated with high ventricular arrhythmic burden and progression towards advanced heart failure.

·       The authors included some patients presenting significant coronary artery disease. This element could potentially influence CPET performance in a significant way.

·       The authors did not provide any information regarding PVC morphology. This element represents a major limitation of this work. PVCs presenting specific features (such as prolonged QRS duration, pseudo delta waves, superior axis or right bundle branch-like morphology) have been associated with specific malignant aetiologies and  worse outcomes.

Comments on the Quality of English Language

I would suggest the authors to perform English proofreading of their manuscript.

Author Response

(The authors gave the same response as above.)

Reviewer 3 Report

Comments and Suggestions for Authors

In this observational study, the authors retrospectively analyzed CPET data of 94 patients with frequent premature ventricular beats (PVC) and compared this with data from 98 control participants. As the main finding, it turned out that cardiorespiratory capacity was lower in patients with frequent PVCs, although there was no relevant difference among subgroups based on the frequency of PVCs.

I have the following remarks:

-        Please make sure that all abbreviations are explained in the text when used for the first time (e.g. CPET in abstract etc.).

-        The authors sometimes use the term “PVC” in situations where they mean the “group of patients with PVC” (see abstract; line 12); please correct

-        Please describe the statistical process of “matching” the groups

-        Paragraph 3.2.: please only give the weight adjusted data (and remove the others)

-        Fig. 1: patient flow is not clear to me; if in the first step, all patients without CPET were removed, so how can it be that there were patients left who were “unwilling to complete CPET” who were then removed in a second step?

-        Fig. 1 suggests that there was no “matching” of patients between the groups but that these groups were rather generated based on the Holter findings; please explain and correct

-        Remove Fig. 2 (un-adjusted data)

-        Fig. 4: please use the same y-axis as in Fig. 3

-        Tabl. 1: some inconclusive data; PVC group: 29 pat. with CAD; however, sum of 1/2/3 vessel disease is only 9; please state in what percentage of patients CAG were available or otherwise explain this finding

-        Tab. 2: high mean value of E/E’ in both groups; please comment

-        Given the fact that in the PVC group there were 53 patients with “PVC<10000/d” (tab. 4) what was the threshold of PVC per day to distinguish between “PVC” and “control group” (assuming that also in the control group some PVC will have occurred)

-        Discussion: “…this is the first report to evaluate the cardiorespiratory capacity induced by PVCs using CPET in adults”; the nature of this study does not allow to state a causal relationship between the findings; please tone down

-        Discussion: the authors state that differences of NTproBNP were (to a certain extent) caused by differences in renal function; so please give data on eGFR in the corresponding table 1

-        Discussion: “We believe that the correlation between the burden of PVCs and important indexes, which showed a negative trend but no statistical significance, could be ascribed to heart rate variability…” Why? Please explain.

-        Conclusion: retrospective study; please tone down your statement accordingly

Author Response

(The authors gave the same response as above.)

Round 2

Reviewer 1 Report

Comments and Suggestions for Authors

I thank the authors for correcting the points I raised.

I have no further corrections to make.

Author Response

Thanks very much again for reviewing our revised manuscript and this article has greatly improved with your help.

Reviewer 3 Report

Comments and Suggestions for Authors

Hello, still some minor comments:

- SD for NTproBNP is missing in table. 1

- number of decimal points not used uniformly in tab. 3 and 4 

Author Response

Thanks again for your kind comments. 

Q1: SD for NTproBNP is missing in the table 1

A1: In fact, the value of NTproBNP was not normally distributed. So we expressed the value by median. And we tagged its p-value in the table 1.

Q2: number of decimal points not used uniformly in tab. 3 and 4

A2: Thank you for your professional advice. We are so sorry to neglect these details. We have corrected the number of decimal points according to your suggestion. 

Very thanks for your comments and your suggestion does help us and improve our article a lot.